# Improved Formazan Dissolution for Bacterial MTT Assay

Ludmil Benov[a]

aDepartment of Biochemistry, Faculty of Medicine, Kuwait University, Kuwait City, Kuwait

**ABSTRACT** The MTT assay, based on the enzymatic reduction of the water-soluble, yellowish tetrazolium salt 3-(4,5-dimethylthiazol)-2,5-diphenyl-tetrazolium bromide (MTT) to purple formazan, is commonly used for assessment of cell viability and proliferation. Accurate performance by the MTT assay depends on complete solubilization of cells and formazan and stability of the colored solution. Comparison of different solubilization solutions revealed that dimethylformamide (DMF) and dimethyl sulfoxide (DMSO), buffered with ammonia buffer, pH 10, and containing 5% SDS, produced the best results. These two solvents provided rapid and complete solubilization of formazan and cells, with minimal background absorbance at 700 nm, good reproducibility (low interassay coefficient of variation), high sensitivity, and color stability for at least 24 h. A linear relationship between viable-cell number and formazan absorbance was preserved for cell densities up to $\sim 1 \times 10^9$ cells/mL for Gram-negative and Gram-positive microorganisms. Since MTT can be reduced by medium components in the absence of cells, blanks containing all medium components but no cells should be run simultaneously. Measurements at two wavelengths, one corresponding to absorption peak of formazan (570 nm) and a background absorbance far from the peak (700 nm), are necessary to avoid artifacts due to incomplete solubilization and turbidity.

**IMPORTANCE** Reduction of the water-soluble tetrazolium salt 3-(4,5-dimethylthiazol)-2,5 diphenyl-tetrazolium bromide (MTT) to purple, water-insoluble formazan is commonly used for assessment of cell viability and proliferation. Spectrophotometric detection of formazan requires its solubilization. The solubilization solvent has a strong influence on data acquisition and often introduces artifacts, leading to misreading of results. This study offers a choice of solvents that minimize solubilization artifacts when the MTT test is applied to microbiological cultures.

**KEYWORDS** bacterial viability, formazan dissolution, MTT assay, solubilization

Enzymatic reduction of the yellowish, water-soluble tetrazolium salt 3-(4,5-dimethylthiazol)-2,5 diphenyl-tetrazolium bromide (MTT) to purple, water-insoluble formazan is among the most commonly used assays for cell viability and proliferation (1, 2). Among its advantages are simultaneous measurement of multiple samples using 96-well-plate readers and automation (3). The MTT assay can be divided into two stages. The first stage is conversion of MTT to colored formazan by metabolically active cells. Formazan has low solubility in water and forms complexes with cells (4). To make spectrophotometric measurements possible, in the second stage, formazan crystals need to be dissolved. Both stages are prone to artifacts. Numerous studies have investigated the influence of various factors on the reduction of MTT to formazan (4–12), but the effect of the solvent as a source of error is much less studied. Among the tested solvents (reviewed in reference 13) were acidified isopropanol (2, 14), ethanol (2), dimethyl sulfoxide (DMSO) (15), ammonia-DMSO (16), 20% sodium dodecyl sulfate (SDS) in 50% dimethylformamide (DMF) (17, 18), 20% SDS in 0.2 M HCl (19), etc. Among the main sources of inaccuracies at the second stage of the MTT assay are incomplete formazan dissolution, pH-dependent shift of the position and height of the formazan absorption peak, precipitation of medium and cell

Address correspondence to ludmil.benov@ku.edu.kw.

The authors declare no conflict of interest.

components, leading to turbidity, and low stability of formazan solutions (13, 14). Selecting the proper solvent is more problematic for bacterial cultures. Bacteria are more difficult to solubilize than mammalian cells, and as already mentioned, formazan forms complexes with bacterial cells, which at the second stage of the MTT assay coprecipitate, leading to erroneous results. The dependence of the MTT assay outcome on the dissolution stage stimulates the search for a solvent that would eliminate, if not all, at least some of the listed sources of error. Theoretically, an ideal solvent for the MTT assay should dissolve formazan completely and fast without forming precipitates and should liberate formazan from complexes with cells; the color should be stable for hours after dissolution without a shift of the position of the formazan absorption peak and without changes in optical properties.

In an attempt to improve the solubilization of formazan in the MTT assay applied to microbiological samples, in this study SDS was added at various concentrations to solutions containing EDTA, buffered DMF, or buffered DMSO, and solubilization of samples containing formazan, produced by Gram-negative and Gram-positive bacteria, was performed. Results demonstrated that 5% SDS solution in DMSO or DMF buffered with ammonia buffer completely solubilized formazan and both Gram-negative and Gram-positive bacteria, producing a clear solution without precipitates or aggregates and giving stable readings even 24 h after solubilization.

## RESULTS AND DISCUSSION

In addition to dissolving formazan crystals, a solvent used for bacterial MTT assay must completely solubilize bacterial cells to avoid coprecipitation, trapping of formazan crystals, and turbidity due to insolubilized cells and cellular fragments. Cell-solubilizing solutions usually contain SDS, sometimes combined with EDTA. On the other hand, DMSO and DMF are preferred solvents for dissolving formazan (15, 18, 20). In order to test the suitability of detergents and organic solvents for application in the MTT assay for bacteria, solubilization solutions were prepared by combining DMSO and DMF with SDS at various concentrations. Since pH affects the absorption spectrum of formazan (21), and stability of formazan solution is low at acidic pH (20, 22), the organic solutions were buffered with ammonia buffer, pH 10.0. Using a buffer with sufficient buffering capacity prevents shifts of the position of the formazan peak, precipitation, and fluctuations of absorbance due to medium components and supplements and stabilizes formazan solution. Due to limited solubility of SDS, the highest SDS concentrations that were achieved in buffered DMF and DMSO were 10% and 15%, respectively.

The aim of our initial experiments was to test the solubility of pure formazan in selected solvents. Optical density (OD) was measured at the formazan peak, determined by recording the absorption spectrum for each solvent and at 700 nm, away from the specific formazan absorption. Absorption spectra demonstrated that formazan absorption has a maximum at 570 nm in all tested solvents except in ammonia-DMSO, where the maximum was at 550 nm, as reported by Wang et al. (16).

Figure 1 shows the relationship between formazan concertation and absorbance at formazan peak minus background OD at 700 nm. Results show that in the absence of organic solvent, formazan crystals have poor solubility in ammonia buffer containing EDTA and SDS (Fig. 1A). Precipitate could be seen in wells containing more than 12.5 $\mu$g formazan (Fig. S2, row a). The presence of undissolved formazan resulted in high absorbance at 700 nm and distorted shape of formazan absorption peak in the recorded spectra. As a result, $OD_{570–700}$ did not increase with the increase of formazan concertation (Fig. 1A).

Dissolving formazan in 5% SDS-buffered DMF produced better results, giving an almost linear increase of $OD_{570–700}$ at formazan concentrations up to 250 $\mu$g/mL (25 $\mu$g/well) (Fig. 1B). Above this concentration, absorption reached values of ~3.0, which makes measurements unreliable, and formazan crystals remained undissolved. Undissolved, aggregated formazan could be observed in the wells at formazan concentrations of 37.5 and 50 $\mu$g/well (Fig. S2, row b). Due to undissolved formazan aggregates, the nonspecific absorbance at 700 nm

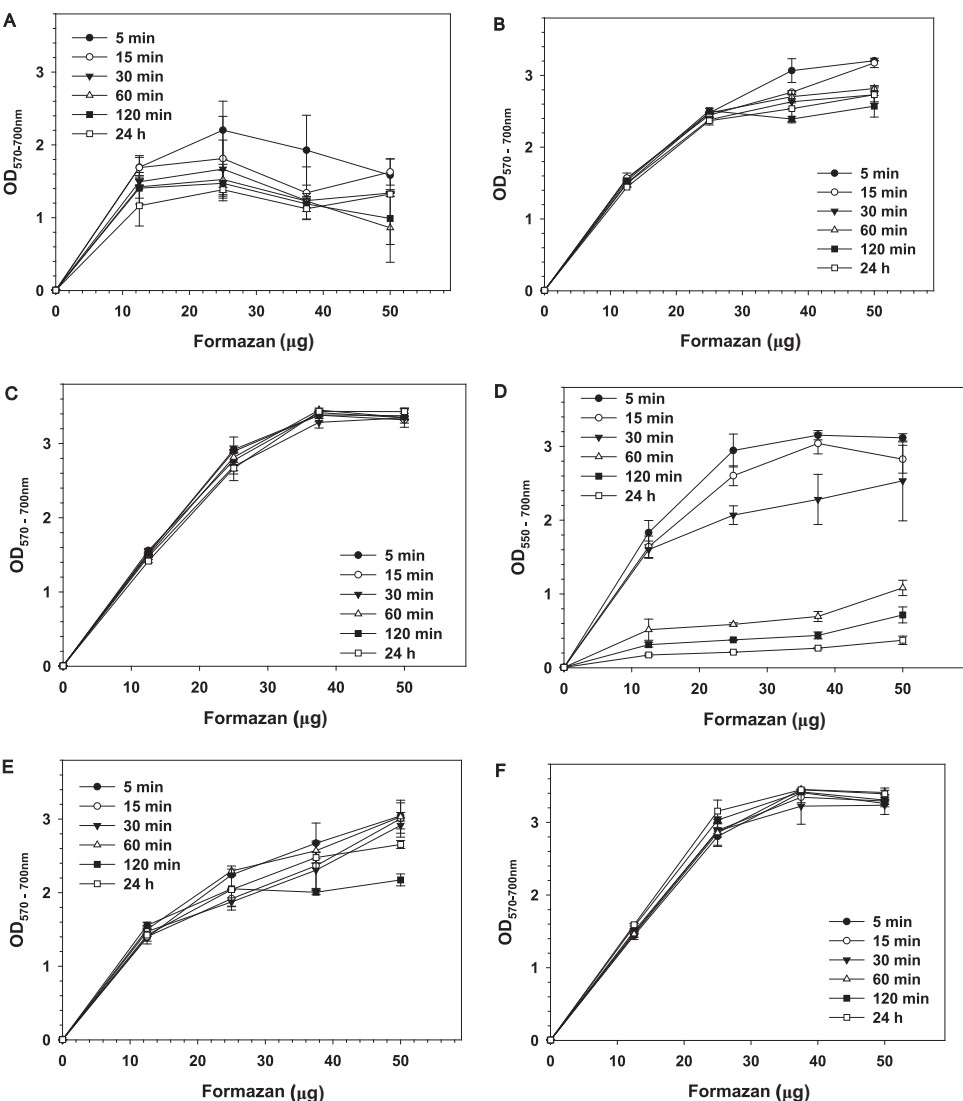

**FIG 1** Effect of solvents on the relationship between formazan concertation and optical density. Solvents: SDS-EDTA (A), 5% SDS-buffered DMF (B), 10% SDS-buffered DMF (C), ammonia-DMSO (D), 5% SDS-buffered DMSO (E), and 10% SDS-buffered DMSO (F). Results are means and SD from three independent experiments. Formazan crystals were dispersed in phosphate-buffered saline (PBS) and serially diluted with PBS with constant stirring. Hundred-microliter aliquots were transferred to triplicate wells in 96-well plates, and 100 $\mu$L of solvent mixtures was added to each well. Plates were kept on a shaker at 200 rpm, and readings were taken at the indicated times.

increased at the highest formazan concentrations. Extending the time of incubation with the solvent from 5 min to 24 h resulted in fluctuations of $OD_{570}$. Similar fluctuations of optical density were observed at 700 nm, which suggests that they resulted from a change of the position of formazan aggregates relative to the light beam of the plate reader.

Doubling the SDS concentration in buffered DMF improved formazan solubilization (Fig. 1C). No undissolved crystals were observed in the wells even at the highest formazan concentration (Fig. S2, row d), which resulted in low background OD at 700 nm. No time-dependent changes in OD were observed when readings were taken from 5 min to 24 h after the addition of the solvent (Fig. 1C). These results indicate that the formazan crystals were completely dissolved within 5 min of incubation and that the formazan solution was stable for at least 24 h.

It should be mentioned that high concentrations of formazan were used only to test the capacity of the solvents to dissolve it. Even if such concentrations could be achieved in some MTT protocols, the absorbance will be too high to allow correct measurements.

When formazan was dissolved in ammonia-DMSO (16) and readings were taken up to 15 min after addition of the solvent, OD concentration dependence was close to that obtained with 5% SDS-buffered DMF (compare panels B and D in Fig. 1). Increasing the time of solubilization, however, resulted in a decrease in absorbance (Fig. 1D). The reason was aggregation of formazan and decrease of its concentration in the solution. This change was seen when wells were compared at different times after addition of the solvent (Fig. S2, row e, and Fig. S3). A time-dependent decrease of OD at 550 nm due to aggregation of formazan was confirmed when absorption spectra were recorded. Figure S4 shows that even for the lowest formazan concentration tested (12.5 $\mu$g/well), the height of the absorption peak decreased time-dependently while the background OD at 700 nm increased. Undissolved formazan caused big variations of OD among wells having the same content.

Replacement of ammonia-DMSO with 5% SDS in buffered DMSO resulted in better solubilization of formazan. No significant time-dependent change of OD was observed for formazan concentrations up to 25 $\mu$g/well when readings were taken 5, 15 30, 60, and 120 min and 24 h after the addition of the solvent (Fig. 1E). At concentrations above 25 $\mu$g/well, formazan did not dissolve well, which led to increase of the background OD at 700 nm.

Increasing the SDS concentration in buffered DMSO to 10% improved formazan solubility (Fig. 1F). No precipitation was observed even at the highest formazan concentrations. Further increase of SDS concentration to 15% did not produce noticeable improvement. At room temperature, however, DMSO containing SDS at concentrations above 5% crystallized, trapping formazan, and samples had to be warmed to obtain correct measurements.

Results presented so far show that buffering of the two tested organic solvents combined with SDS provided good solubilization of pure formazan at concentrations up to 25 mg/mL and that the formazan solution is stable for at least 24 h. In real assays, however, the solvents must solubilize bacterial cells and release trapped formazan, without causing turbidity.

Subsequent experiments aimed to obtain preliminary data about the ability of solvents to dissolve formazan and cells when the MTT test was performed on bacterial cultures. For initial testing of solvents, *E. coli* cultures with densities of $7 \times 10^8$, $12 \times 10^8$, and $70 \times 10^8$ cells/mL were used. Hundred-microliter samples were incubated with MTT for 30 min in 96-well plates at 37°C and 200 rpm. At the end of the incubation period, selected solvents were added, the plates were kept on a shaker at 200 rpm, and readings were taken 1 and 2 h after the addition of the solvents.

The results of these experiments demonstrated that (i) irrespective of its good cell-solubilizing efficacy, SDS-EDTA did not perform well in the MTT assay for bacteria, and (ii) increasing the concentration of SDS above 5% in the mixtures with buffered organic solvents (DMF and DMSO) did not improve the outcome of the MTT assay but, depending on the temperature, led to crystallization.

Based on these observations, the following solvents were selected for further testing: 5% SDS in buffered DMF, 5% SDS in buffered DMSO, and ammonia-DMSO (16). These solvents were initially tested on serially diluted *E. coli* suspensions subjected to the MTT assay. To check the stability of the formazan solution, readings were taken 1, 2, and 24 h after the addition of the solvent.

When 5% SDS in buffered DMF (Fig. 2A) or 5% SDS in buffered DMSO (Fig. 2B) was used as the solvent, the OD did not change significantly with time. Linear regression analysis of data obtained with these solvents demonstrated that the OD for 5% SDS-buffered DMF and 5% SDS-buffered DMSO proportionally increased with the increase of cell number and that linear regression lines overlap (Fig. 2D).

Different results were obtained when ammonia-DMSO was used for solubilization (Fig. 3C). After 1 h of incubation, the maximal OD obtained for the highest cell number was only 0.84, and the OD dropped to 0.54 at the second hour. With this solvent, absorbance did not increase proportionally to cell number, error bars were large, the background OD at 700 nm was high, and color intensity decreased with time.

The reason for the large error bars became clear when spectra of wells with the same content were compared (Fig. S5A). The cause of the large differences among wells with the

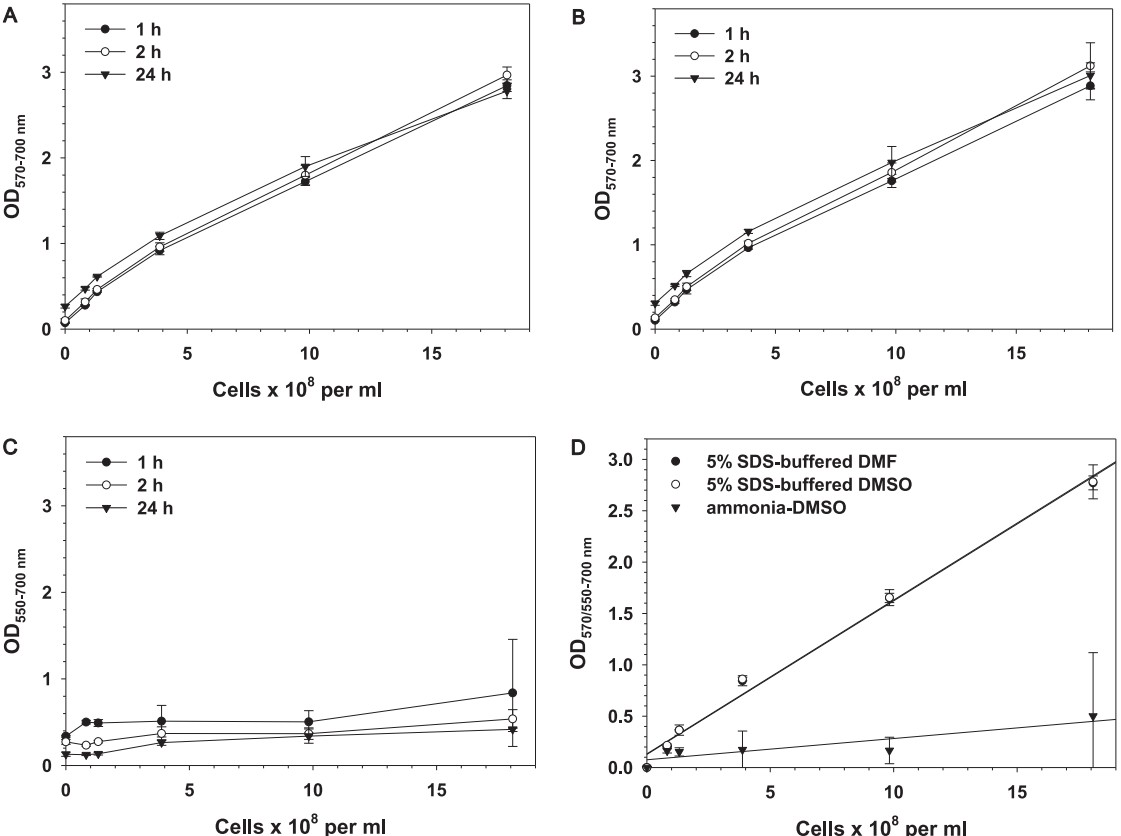

**FIG 2** (A to C) Relationship between *E. coli* cell number and optical density. Solvents: 5% SDS-buffered DMF (A), 5% SDS-buffered DMSO (B), and ammonia-DMSO (C). (D) Linear regression analysis of results obtained by the three tested solvents after 1 h of incubation (OD at the formazan peak, 570 nm for SDS-DMSO and SDS-DMF and 550 nm for ammonia-DMSO, minus the background absorbance at 700 nm is shown). Results are means and SD from three independent experiments. Overnight *E. coli* cultures were serially diluted in LB medium, and 100-$\mu$L aliquots were incubated with MTT solution for 30 min in 96-well plates at 37°C and 200 rpm. At the end of the incubation period, 100 $\mu$L of a solvent mixture was added to each well, and plates were kept on a shaker at 200 rpm. Readings were taken 1, 2, and 24 h after the addition of the solvents.

same content was precipitate that could be observed with naked eye (Fig. S5B and C). As a consequence of formazan aggregation and precipitation, the solution was unstable, and the formazan-specific OD decreased while background $OD_{700}$ increased with time.

The difference between results obtained with ammonia-DMSO and 5% SDS-buffered DMF or 5% SDS-buffered DMSO is evident in Fig. 3D. The two SDS-containing solvents demonstrated higher and identical OD at each cell number, which indicates higher sensitivity and ability to distinguish close cell numbers. Slopes of regression analysis lines show that in contrast to the SDS-containing solvents, ammonia-DMSO provided lower sensitivity. The good linear relationship between cell number and OD for the SDS-containing buffered organic solvents was furthered confirmed by the high coefficient of determination, $R^2$ (Table S1).

The interassay coefficient of variation (CV), which reflects the precision/repeatability of data obtained by independent experiments, is presented in Fig. S7A. The CV for the two SDS-containing solvents is below the recommended maximal value of 15% but is much higher for ammonia-DMSO.

Experiments with a Gram-negative microorganism, *Escherichia coli*, demonstrated that buffered DMF or DMSO containing 5% SDS, efficiently solubilizes formazan and bacteria. It is not certain, however, if these solvents would solubilize Gram-positive bacteria well, due to differences between Gram-negative and Gram-positive microorganisms in content and structure of the cell envelope. To test the usefulness of these solvents for an MTT assay performed on a Gram-positive microorganism, experiments with *Staphylococcus aureus* were carried out.

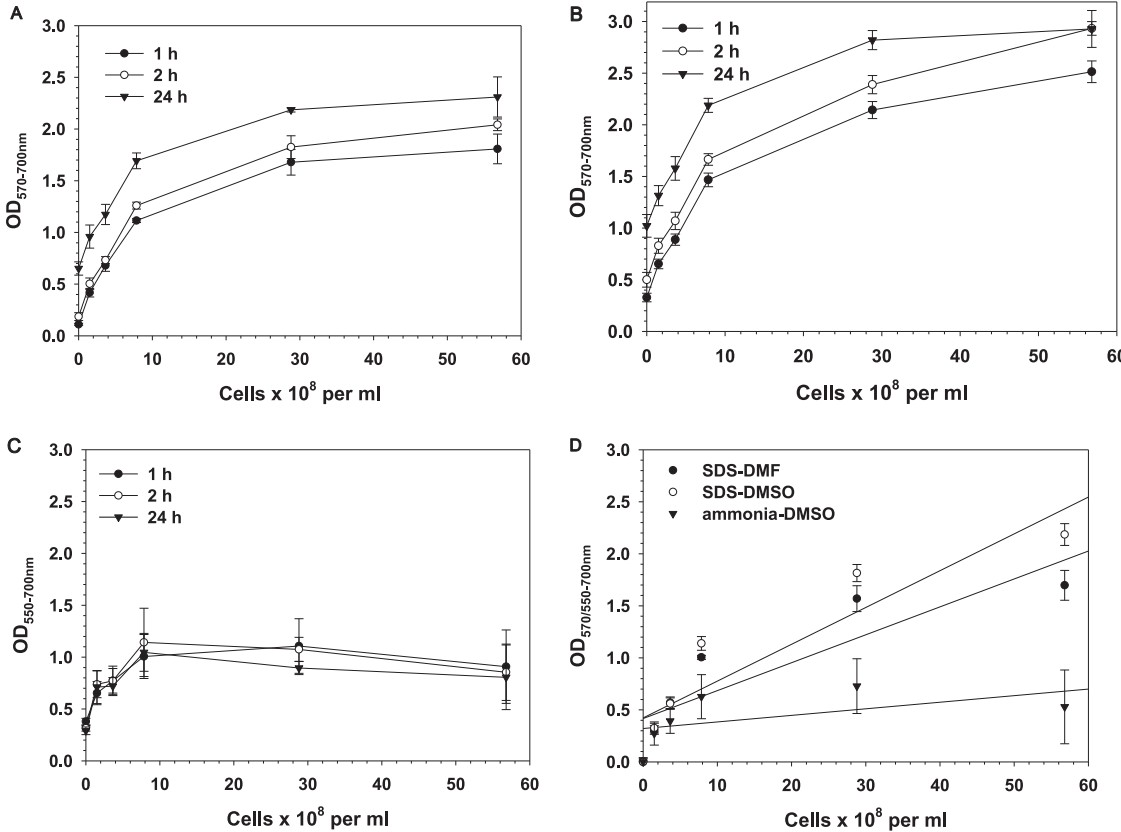

**FIG 3** (A to C) Relationship between *S. aureus* cell number and optical density. Solvents: 5% SDS-buffered DMF (A), 5% SDS-buffered DMSO (B), and ammonia-DMSO (C). (D) Linear regression analysis of results obtained by the three tested solvents after 1 h of incubation (OD at the formazan peak, 570 nm for SDS-DMSO and SDS-DMF and 550 nm for ammonia-DMSO, minus the background absorbance at 700 nm is shown). Results are means and SD from three independent experiments. Overnight *S. aureus* cultures were serially diluted in brain heart infusion (BHI) medium, and 100-$\mu$L aliquots were incubated with MTT solution for 30 min in 96-well plates at 37°C and 200 rpm. At the end of the incubation period, 100 $\mu$L of a solvent mixture was added to each well, and plates were kept on a shaker at 200 rpm. Readings were taken 1, 2, and 24 h after the addition of the solvents.

Figure 3A shows linear increase in OD with the increase in cell density up to $\sim$10$^9$ cell/mL when 5% SDS in buffered DMF was used as a solubilizing solvent. Increasing cell density above 10$^9$ cell/mL resulted in an OD of >2.00 and loss of linearity. The background absorbance (OD$_{700}$) remained lower (<0.3) even at the highest cell densities, which indicates good solubilization, without interference by turbidity or nonsolubilized formazan.

Similar results were obtained when 5% SDS in buffered DMSO was tested as a solubilizing solvent (Fig. 3B).

The outcome was different when ammonia-DMSO was used as a solubilizing solvent. Data presented in Fig. 3C show a lack of correlation between cell number and OD and large error bars. The background absorbance at 700 nm increased to values of 2.00 and above, which resulted in a distorted relationship between cell number and formazan-specific absorbance. As mentioned above, the reason was precipitation and aggregation of formazan. Figure S6A shows wells of a 96-well plate after 2 h of incubation, and Fig. S6B displays spectra of triplicate wells containing 8 $\times$ 10$^8$ cells/mL subjected to the MTT assay, 2 h after addition of ammonia-DMSO. For comparison, spectra of triplicate wells containing the same number of cells and treated the same way but solubilized with 5% SDS-buffered DMSO are shown in Fig. S6C.

Results presented in Fig. 3 show that OD increased with time of incubation with the solvents. Such an increase was due to nonenzymatic reduction of MTT to formazan, because it occurred in blanks containing BHI medium and no cells. When the OD of the blank was deducted, no time-dependent change of OD was observed, which indicates that after solubilization with either 5% SDS-buffered DMF or 5% SDS-buffered DMSO, the color is stable for at least 24 h.

Regression analysis of data obtained when the MTT assay was applied to *S. aureus* is presented in Fig. 3D. It shows that dissolution of formazan by 5% SDS-buffered DMF and 5% SDS-buffered DMSO produced comparable results, but at higher cell numbers, 5% SDS-buffered DMSO gave a slightly higher OD, which can be considered an indication of higher sensitivity. Values of $R^2$ displayed in Table S1 were lower than the $R^2$ values obtained for *E. coli*. The reason can be found in the fact that for *S. aureus*, a linear increase in OD with increasing cell numbers was observed only up to $\sim 1 \times 10^9$ cells/mL. Since linearity in that range was observed with all tested solvents, it can be concluded that above this cell density, formazan production reaches saturation; i.e., it was limited not by the solvent but by the cellular conversion of MTT to formazan. Regression analysis for OD values obtained with cell densities up to $1 \times 10^9$ cells/mL produced $R^2$ values close to those obtained for *E. coli* (Table S1). Again, ammonia-DMSO gave the lowest $R^2$, and the interassay CV for this solvent was far above the 15% recommended maximal value (Fig. S7B).

In summary, the ideal solvent for solubilization of formazan and cells for a microbiological MTT assay should meet the following requirements:

- Fast and complete solubilization of formazan and cells without turbidity and formazan crystallization
- High specific absorbance at the formazan peak
- Minimal background absorbance ($OD_{700}$)
- Minimal interference by media and additives
- Stability of the formazan solution without shift of the position of the formazan peak or decrease of its height
- Linear relationship between viable-cell number and formazan optical absorbance
- Good reproducibility and minimal variability (low coefficient of variation)

Among the tested solvents, only two, 5% SDS-buffered DMF and 5% SDS-buffered DMSO, demonstrated properties close to these requirements. These two solvents provided rapid and complete solubilization of formazan and cells, with minimal background absorbance at 700 nm, good reproducibility, high sensitivity, and color stability for at least 24 h. The linear relationship between viable-cell number and formazan absorbance was preserved for cell densities up to $\sim 1 \times 10^9$ cells/mL for Gram-negative and Gram-positive microorganisms when the MTT protocol described in Materials and Methods was followed. The fact that part of MTT is reduced by the medium in the absence of cells indicates that blanks containing all medium components but no cells should be run simultaneously. Measurements at two wavelengths, one specific for the formazan at its absorption peak (570 nm) and a background absorbance far from the peak (700 nm), should be performed to avoid artifacts due to incomplete solubilization, turbidity, and other interfering factors.

## MATERIALS AND METHODS

**Preparation of formazan.** Formazan was prepared by adding 25 mg of 3-(4,5-dimethylthiazol-2-yl)-2,5-diphenyl-tetrazolium bromide (MTT) (Sigma-Aldrich) per mL of sodium ascorbate solution (150 mg sodium ascorbate per mL). The mixture was stirred for 5 min, and formazan crystals were isolated by filtration and thoroughly washed with distilled water. Pure formazan was dried under vacuum until a constant weight was reached and was stored in tightly closed containers in the dark. For experiments, crystals were weighed and dispersed in phosphate-buffered saline (PBS). Aliquots were serially diluted with constant stirring, and 100-$\mu$L portions were added to triplicate wells in 96-well plates, followed by the addition of 100 $\mu$L of the respective solvent.

**Strains and growth conditions.** The following Gram-negative and Gram-positive strains were used in this study: *Escherichia coli* strain GC4468 (F$^-$ $\Delta lacU169$ *rpsL*) provided by D. Touati (23) and *Staphylococcus aureus* strain ATCC 25923 (24). Cells were inoculated from freezer stocks in Luria-Bertani (LB) (*E. coli*) or in brain heart infusion (BHI) broth (BD Biosciences, USA) (*S. aureus*) and were grown overnight (14 to 16 h in a shaking water bath at 200 rpm and 37°C). For experiments, the overnight cultures were serially diluted in LB (*E. coli*) or BHI (*S. aureus*) medium to cell densities in the ranges of $7 \times 10^8$ to $7 \times 10^9$ cells/mL (*E. coli*) and $1.5 \times 10^8$ to $5.7 \times 10^9$ cells/mL (*S. aureus*). Cell numbers were determined by plating on agar plates and counting colonies. Simultaneously, the optical density at 600 nm was measured for each dilution and was used for determining cell densities in repeated experiments. These serially diluted cultures were subjected to the MTT assay without further manipulations. Hundred-microliter aliquots were added to triplicate wells in 96-well plates, and the MTT assay was performed as described below.

**MTT assay.** The MTT reagent was prepared by dissolving 25 mg of MTT in 5 mL PBS. Ten microliters of MTT reagent was added to all wells, and plates were incubated for 30 min on a thermostatic shaker at 37°C and 200 rpm in the dark. It has been established that MTT reduction by Gram-positive and Gram-negative microorganisms occurs within the first 15 to 30 min, and extension of the time of incubation does not increase the formazan (25). After 30 min, 100-$\mu$L aliquots of solvents were added to each well, and plates were kept on a shaker at 200 rpm. At definite time intervals, absorption spectra were recorded, and absorbance at the maximum of the formazan peak and at a reference wavelength (700 nm) was measured using a microplate reader (CLARIOstar; BMG Labtech, Inc., USA). Data were analyzed using MARS data analysis software provided by the manufacturer. In all experiments, controls containing reagents but no cells were run in parallel.

A reference wavelength of 700 nm was selected as the wavelength giving low absorbance outside the formazan maximum (26) (Fig. S1). In all experiments, readings were taken at dual wavelengths (570 nm and 700 nm for all solvents, except that 550 nm and 700 nm were used for ammonia DMSO) to eliminate nonspecific absorbance resulting from turbidity, crystallization, etc.

**Solubilization solutions.** Selection of solubilizing solutions was based on literature data and preliminary experiments. Solution mixtures that led to turbidity, caused precipitation of medium components and cells, and decreased the sensitivity of the MTT assay were not used. The following solvents were tested for solubilization of formazan and cells: (i) 5% SDS in ammonia buffer, pH 10.00, containing 10 mM EDTA, referred to as SDS-EDTA; (ii) 5% and 10% SDS in ammonia buffer containing 87.5% DMF, referred to as 5% SDS-buffered DMF and 10% SDS-buffered DMF; (iii) 880 mM ammonia with 94% DMSO (16), referred to as ammonia-DMSO; and (iv) 5%, 10%, and 15% SDS in ammonia buffer, pH 10.00, containing 87.5% DMSO, referred to as 5% SDS-buffered DMSO, 10% SDS-buffered DMSO, and 15% SDS-buffered DMSO, respectively.

**Preparation of ammonia buffer, pH 10.00.** Ammonia buffer was prepared by dissolving 5.4 g ammonium chloride in 20 mL distilled water and adding 35 mL of 10 M ammonia solution. The pH was adjusted to 10.00, and the buffer was diluted to 100 mL with deionized water.

**Data analysis.** Each experiment was repeated at least twice with no fewer than three replicates. One-way analysis of variance (ANOVA) was performed using SigmaPlot version 11.0, and $P$ values of $<0.05$ were accepted as statistically significant. Data are presented as means and standard deviations (SD).

**(i) Coefficient of variation.** The interassay CV was determined by dividing the SD by the mean of data obtained by repetition of independent experiments and is presented as a percentage of the SD.

**(ii) Linear regression analysis.** The relationship between formazan-specific OD and cell number was analyzed by linear regression analysis using SigmaPlot version 11.0 software.

**Data availability.** Raw data are available at Open Science Framework (osf.io/edaf6).

## SUPPLEMENTAL MATERIAL

Supplemental material is available online only.

**SUPPLEMENTAL FILE 1**, PDF file, 0.4 MB.

## ACKNOWLEDGMENTS

This work was supported by Kuwait University grant MB01/18.

I am grateful to F. Sequeira for excellent technical assistance and to Danielle Touati (Institute Jacques Monod, CNRS, University Paris, France) and E. Udo (Kuwait University, Kuwait) for providing bacterial strains.

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
