## [Reviewer comments · Microbiology Spectrum]

Microbiology Spectrum

Improved formazan dissolution for bacterial MTT assay

Ludmil Benov

Corresponding Author(s): Ludmil Benov, Kuwait University, Faculty of Medicine

Review Timeline:

Submission Date:	September 22, 2021
Editorial Decision:	October 28, 2021
Revision Received:	November 25, 2021
Accepted:	November 30, 2021

Editor: Jeffrey Gralnick

Reviewer(s): Disclosure of reviewer identity is with reference to reviewer comments included in decision letter(s). The following individuals involved in review of your submission have agreed to reveal their identity: Yeong Ji Oh (Reviewer #1); Steven C. Olsen (Reviewer #2)

Transaction Report:

DOI: <https://doi.org/10.1128/spectrum.01637-21>

October 28, 2021

Dr. Ludmil T. Benov
Kuwait University, Faculty of Medicine
Biochemistry
POBox 24923 Safat
Kuwait 13110
Kuwait

Re: Spectrum01637-21 (Improved formazan dissolution for bacterial MTT assay)

Dear Dr. Ludmil T. Benov:

The reviewers found merit in your study, but suggest that it needs to be subjected to major modifications before it is ready for publication. In particular, a statistical analysis of the results would be needed to draw scientifically sound conclusions on the suitability of the different solvents is missing. In addition, the manuscript needs to be shortened and rewritten to increase clarity.

Thank you for submitting your manuscript to Microbiology Spectrum. When submitting the revised version of your paper, please provide (1) point-by-point responses to the issues raised by the reviewers as file type "Response to Reviewers," not in your cover letter, and (2) a PDF file that indicates the changes from the original submission (by highlighting or underlining the changes) as file type "Marked Up Manuscript - For Review Only". Please use this link to submit your revised manuscript - we strongly recommend that you submit your paper within the next 60 days or reach out to me. Detailed information on submitting your revised paper are below.

Link Not Available

Sincerely,

Sophia Johler

Journals Department
Reviewer comments:

Reviewer #1 (Comments for the Author):

This research paper is good and could provide that 5 % SDS in DMF or in DMSO are suitable solvents for solubilizing bacterial formazan crystal using MTT colorimetric method. However, there are some overlapping information with the literature for some part of the paper.

1. Discuss the originality (contribution, addition of knowledge to scientific literature or field) of the manuscript.
2. The abstract of the MS is not well documented to represent the whole manuscript objectives. Thus, it should be rewritten again in order to represent the paper sufficiently.
3. It will be useful to convert figure data to one table since there are not table.
4. Statistical analysis have to be discussed comprehensively in the results and discussion section as well.
5. Line 43: Additional experiments on misreading results caused by interference factors (media, reagents, samples) are needed.
6. Clearly identify the strengths and weaknesses of the method described in the manuscript.

7. Discussion of the paper has to be modified using with some recent references and some excessive explanations have to be removed.
8. Excessive supplementary data should be summarized and removed.
9. Line 97: Methods (inoculation, centrifugation,,,) and the sequence of experiments should be described in more detail.

Reviewer #2 (Comments for the Author):

In this manuscript the authors explore new methods for solubilization of formazan formed from MTT in an effort to improve the accuracy of laboratory procedures using MTT as an indication of viability.

Major comments:

- 1) The authors measured absorbance at 570 and 600 with the 600 reading serving as a background measurement outside of the maximum of formazan peak absorbance. However, all graphs depict only the AB570 measurement and do not adjust for the 600 nm background measurement. The authors even recommend in the conclusion on page 19 that measurements at two wavelengths be conducted (570 and 600 nm). For this reason, I believe the presented data should include deduction of the background.
2. The experiment measuring MTT conversion in culture of *Staphylococcus aureus* differs in that the graphs do not show measurements at 1, 2 and 24 hr but only 1 measurement. Text and figure legend do not provide the information on the incubation time for this set of data.
3. One might argue whether statistical analysis would be beneficial or not for this set of data. However, particularly in Fig 4, I believe there would be benefits for statistical analysis comparing OD with the different solubilization solutions. For example was OD for the ammonia-DMSO greater than the other 2 solutions? Because of the nature of the curve generated for the assay it may be difficult to develop a regression analysis that would allow comparison of different treatments. However, I would suggest the authors consider the possibility of doing this type of analysis.
3. Descriptions of experiments presented in Figures 1, 2, and 3 get rather tedious. Would it be possible to prepare a graph presenting only 1 incubation time for each compound and statistically compare the OD at different concentrations (of formazan or bacteria). The presentation of the data in the figure could be more comparative, and information on stability over time in incubation, turbidity, and other data could then be discussed in the text. This suggestion is supported by the strong correlation in curves across incubations This might make the data more comprehensible to readers. This would also provide clarity on what constituted the most desired results for each experiments (most likely maximum 570 abs and minimum 600 nm absorbance).
- 4) In the methods on line 190-110, the 570 nm wavelength should be specifically stated. On line 115 I would suggest listing the dual wavelengths in parentheses so that the reader is clear on what wavelengths were used.
- 5) On line 119, I believe "decreed" should be "decreased".
- 6) Lines 257 to 258 are confusing to this reviewer. I'm unclear on how the methods would detect a shift in the formazan peak since the authors were only measuring at 570 nm.
- 7) Lastly, I think the readability of the paper would be improved if the authors articulated what constituted the best solvent for solubilization of formazan. I would think it would be maximal absorbance at 570 nm (for a particular formazan or bacterial concentration), minimal variance, and minimal absorbance at 600 nm.

Staff Comments:

Preparing Revision Guidelines

Please return the manuscript within 60 days; if you cannot complete the modification within this time period, please contact me. If you do not wish to modify the manuscript and prefer to submit it to another journal, please notify me of your decision immediately so that the manuscript may be formally withdrawn from consideration by Microbiology Spectrum.

This research paper is good and could provide that 5 % SDS in DMF or in DMSO are suitable solvents for solubilizing bacterial formazan crystal using MTT colorimetric method. However, there are some overlapping information with the literature for some part of the paper.

1. Discuss the originality (contribution, addition of knowledge to scientific literature or field) of the manuscript.
2. The abstract of the MS is not well documented to represent the whole manuscript objectives. Thus, it should be rewritten again in order to represent the paper sufficiently.
3. It will be useful to convert figure data to one table since there are not table.
4. Statistical analysis have to be discussed comprehensively in the results and discussion section as well.
5. Line 43: Additional experiments on misreading results caused by interference factors (media, reagents, samples) are needed.
6. Clearly identify the strengths and weaknesses of the method described in the manuscript.
7. Discussion of the paper has to be modified using with some recent references and some excessive explanations have to be removed.
8. Excessive supplementary data should be summarized and removed.
9. Line 97: Methods (inoculation, centrifugation,,,) and the sequence of experiments should be described in more detail.

Response to reviewers

Reviewer #1 (Comments for the Author):

I am grateful to Reviewer # 1 for the useful suggestions and recommendations for improving this manuscript. All suggestions were taken into consideration and suitable changes have been introduced in the revised version.

This research paper is good and could provide that 5 % SDS in DMF or in DMSO are suitable solvents for solubilizing bacterial formazan crystal using MTT colorimetric method. However, there are some overlapping information with the literature for some part of the paper.

1. Discuss the originality (contribution, addition of knowledge to scientific literature or field) of the manuscript.

- This has been done by describing the effect of buffering and addition of SDS to tested organic solvents (Results & Discussion section)

2. The abstract of the MS is not well documented to represent the whole manuscript objectives. Thus, it should be rewritten again in order to represent the paper sufficiently.

- The abstract has been rewritten

3. It will be useful to convert figure data to one table since there are not table.

- Due to the character of the data, attempt to convert figures to a table, resulted in a confusing set of numbers. A table, however, has been added, summarizing the results of the linear regression analysis (Suppl. Table 1).

4. Statistical analysis have to be discussed comprehensively in the results and discussion section as well.

- Additional statistical analyses were performed (linear regression, and coefficient of variation), results were presented as Figs. 2 D and 3 D, Suppl. Fig. S7, and Suppl. Table 1. Details were given in the methodological section, and results were discussed in the Results and Discussion section.

5. Line 43: Additional experiments on misreading results caused by interference factors (media, reagents, samples) are needed.

- Distortion of results caused by interference factors, including precipitation, crystallization, low solubilization, shifts of absorption peaks, and instability of solutions have been demonstrated in the presented figures incorporated in the main text and in the supplementary figures, and have been discussed in the Results and Discussion section.

6. Clearly identify the strengths and weaknesses of the method described in the manuscript.

- The advantages and disadvantages of each tested solvent have been discussed at places where data about the performance of the solvent were presented.

7. Discussion of the paper has to be modified using with some recent references and some excessive explanations have to be removed.

- Thorough literature search produced only two recent publications relevant to the topic of this study and they were added to the references list (Ref. 22 & 26)

8. Excessive supplementary data should be summarized and removed.

- Supplementary material has been shortened and excessive figures have been removed

9. Line 97: Methods (inoculation, centrifugation,,,) and the sequence of experiments should be described in more detail.

- Details about preparation and manipulation of samples have been added to the experiment section

Reviewer #2 (Comments for the Author):

I appreciate the time and efforts spent by Reviewer #2 for improving this manuscript. I am grateful for the criticism and believe all suggestions were properly addressed and suitable corrections were done.

In this manuscript the authors explore new methods for solubilization of formazan formed from MTT in an effort to improve the accuracy of laboratory procedures using MTT as an indication of viability.

Major comments:

1) The authors measured absorbance at 570 and 600 with the 600 reading serving as a background measurement outside of the maximum of formazan peak absorbance. However, all graphs depict only the AB570 measurement and do not adjust for the 600 nm background measurement. The authors even recommend in the conclusion on page 19 that measurements at two wave lengths be conducted (570 and 600 nm). For this reason, I believe the presented data should include deduction of the background.

- **All figures now present OD at the formazan peak (570 nm for all solvents except ammonia DMSO which has a max. at 550 nm) with the background absorbance at 700 nm deducted.**

2. The experiment measuring MTT conversion in culture of Staphylococcus aureus differs in that the graphs do not show measurements at 1, 2 and 24 hr but only 1 measurement. Text and figure legend do not provide the information on the incubation time for this set of data.

- All figures have been replaced and results obtained at 1, 2 and 24 hours have been shown

3. One might argue whether statistical analysis would be beneficial or not for this set of data. However, particularly in Fig 4, I believe there would be benefits for statistical analysis comparing OD with the different solubilization solutions. For example was OD for the ammonia-DMSO greater than the other 2 solutions? Because of the nature of the curve generated for the assay it may be difficult to develop a regression analysis that would allow comparison of different treatments. However, I would suggest the authors consider the possibility of doing this type of analysis.

-Linear regression analysis has been performed and results have been presented in Figs. 2 D, 3 D, and Suppl. Table 1. In addition, inter-assay coefficient of variation has been calculated. Results were presented as Suppl. Fig. S7 and discussed in the Results and Discussion section.

3. Descriptions of experiments presented in Figures 1, 2, and 3 get rather tedious. Would it be possible to prepare a graph presenting only 1 incubation time for each compound and statistically compare the OD at different concentrations (of formazan or bacteria). The presentation of the data in the figure could be more comparative, and information on stability over time in incubation, turbidity, and other data could then be discussed in the text. This suggestion is supported by the strong correlation in curves across incubations. This might make the data more comprehensible to readers. This would also provide clarity on what constituted the most desired results for each experiment (most likely maximum 570 abs and minimum 600 nm absorbance).

- Results obtained at different incubation times are presented to give the reader a direct perception of the stability of formazan in different solvents, to show advantages and disadvantages of the tested solvents, and to demonstrate how artifacts distort the readings.

One set of figures (former Fig. 2, A-F) has been removed and results described in the text. Figs 2 D and 3 D now present regression analysis lines of only one time point (1 hour).

4) In the methods on line 190-110, the 570 nm wavelength should be specifically stated. On line 115 I would suggest listing the dual wavelengths in parentheses so that the reader is clear on what wavelengths were used.

- Formazan absorption maximum wavelengths (570 nm for all solvents except ammonia DMSO, which shows a max. at 550 nm) have been added to the text.

5) On line 119, I believe "decreed" should be "decreased".

- The typo has been corrected

6) Lines 257 to 258 are confusing to this reviewer. I'm unclear on how the methods would detect a shift in the formazan peak since the authors were only measuring at 570 nm.

- A shift of formazan peak has been determined by recording and comparing absorption spectra. Selected spectra have been shown as supplementary figures. The paragraph mentioned above has been modified.

7) Lastly, I think the readability of the paper would be improved if the authors articulated what constituted the best solvent for solubilization of formazan. I would think it would be maximal absorbance at 570 nm (for a particular formazan or bacterial concentration), minimal variance, and minimal absorbance at 600 nm.

- A paragraph has been added (L. 421 – 431) listing the desired properties of a good solvent

November 30, 2021

Dr. Ludmil T. Benov
Kuwait University, Faculty of Medicine
Biochemistry
POBox 24923 Safat
Kuwait 13110
Kuwait

Re: Spectrum01637-21R1 (Improved formazan dissolution for bacterial MTT assay)

Dear Dr. Ludmil T. Benov:

Your manuscript has been accepted, and I am forwarding it to the ASM Journals Department for publication. You will be notified when your proofs are ready to be viewed.

Sincerely,

Jeffrey Gralnick
Editor, Microbiology Spectrum
